# Association of Chocolate Consumption with Hearing Loss and Tinnitus in Middle-Aged People Based on the Korean National Health and Nutrition Examination Survey 2012–2013

**DOI:** 10.3390/nu11040746

**Published:** 2019-03-30

**Authors:** Sang-Yeon Lee, Gucheol Jung, Myoung-jin Jang, Myung-Whan Suh, Jun ho Lee, Seung-Ha Oh, Moo Kyun Park

**Affiliations:** 1Department of Otorhinolaryngology-Head and Neck Surgery, Seoul National University Hospital, Seoul 03080, Korea; maru4843@hanmail.net (S.-Y.L.); drmung@naver.com (M.-W.S.); junlee@snu.ac.kr (J.h.L.); shaoh@snu.ac.kr (S.-H.O.); 2Medical Research Collaborating Center, Seoul National University Hospital, Seoul 03080, Korea; 9E887@snuh.org (G.J.); mjjang2014@naver.com (M.-j.-J.); 3Sensory Organ Research Institute, Seoul National University Medical Research Center, Seoul 03080, Korea

**Keywords:** chocolate, hearing loss, tinnitus, cohort study

## Abstract

Chocolate, which is produced from cocoa, exerts antioxidant and anti-inflammatory effects that ameliorate neurodegenerative diseases. We hypothesized that chocolate consumption would protect against hearing loss and tinnitus. We evaluated the hearing and tinnitus data, as well as the chocolate consumption, of middle-aged participants (40–64 years of age) of the 2012–2013 Korean National Health and Nutrition Examination Survey. All of the subjects underwent a medical interview, physical examination, audiological evaluation, tinnitus questionnaire, and nutrition examination. A total of 3575 subjects 40–64 years of age were enrolled. The rate of any hearing loss (unilateral or bilateral) in the subjects who consumed chocolate (26.78% (338/1262)) was significantly lower than that in those who did not (35.97% (832/2313)) (*p* < 0.001). Chocolate consumption was independently associated with low odds of any hearing loss (adjusted odds ratio = 0.83, 95% confidence interval = 0.70 to 0.98, *p* = 0.03). Moreover, the severity of hearing loss was inversely correlated with the frequency of chocolate consumption. In contrast to chocolate, there was no association between hearing loss and the consumption of sweet products without cocoa. Chocolate consumption was also not associated with tinnitus or tinnitus-related annoyance. Our results suggest that a chocolate-based diet may protect middle-aged people from hearing loss.

## 1. Introduction

Hearing loss, a highly prevalent sensorineural disorder, imposes a major economic and social burden [1]. The incidence of hearing loss is approximately 20% when mild and unilateral hearing losses are included [2]. Hearing loss typically hampers communication and relationships, thereby resulting in social isolation [3]. In addition, hearing loss increases the risk of depression symptoms and deterioration of the quality of life [4]. Hearing loss is an important risk factor for neurodegenerative dementia [5,6]. Auditory rehabilitation using, for example, hearing aids and cochlear implants, restores auditory function, but the protective effect of certain foodstuffs against hearing loss is unclear.

Tinnitus is a common otologic symptom, often called a “phantom sound” as it is a conscious auditory perception without a corresponding physical source [7]. The prevalence of tinnitus ranges from 12% to 30% per 100,000 people [8,9]. Tinnitus can be bothersome and negatively affect sleep, concentration, emotions, and social enjoyment [10]. Tinnitus is associated particularly with hearing loss, and so correction of any cochlear pathology would ameliorate tinnitus and tinnitus-related distress. However, the selection of a treatment for tinnitus is supported by limited evidence [11]. Moreover, neither medical therapy nor dietary supplements substantially reduce tinnitus-related distress [12].

Patients and clinicians have concerns regarding the effect of diet on hearing loss and tinnitus [13,14]. Therefore, efforts to identify foods that protect hearing are needed. The health-promoting effects of cocoa have attracted the attention of researchers, health-conscious consumers, and manufacturers of cocoa products. Chocolate, which is produced from cocoa, is consumed for pleasure worldwide and is a source of health-promoting compounds [15]. Chocolate contains abundant polyphenols, the antioxidant and anti-inflammatory effects of which might protect against audiological impairment. Recent animal studies demonstrated that polyphenols attenuate oxidative stress and inflammation in the cochlea [16,17]. However, no human study has investigated the association between chocolate intake and hearing loss and/or tinnitus.

Based on the therapeutic effect of chocolate [18], we hypothesized that chocolate consumption would decrease the burden of hearing loss and tinnitus. We investigated the effect of chocolate consumption on hearing loss and tinnitus in middle-aged people in a large Korean cohort.

## 2. Materials and Methods

### 2.1. Study Population

This cohort study used data, from the Korean National Health and Nutrition Examination Survey (KNHANES), representative of the health and nutritional status of Koreans. We examined the association of chocolate consumption with hearing loss and tinnitus in participants 40 to 65 years of age. All subjects gave their informed consent for inclusion before they participated in the study. This study was approved by the Institutional Review Board of Seoul National University Hospital (1902-046-1008). 

A total of 16,076 individuals participated in the 2012–2013 KNHANES. Subjects with external or middle ear pathologies, a history of congenital hearing loss, or retrocochlear lesions were excluded from the study. Of the remaining 5673 middle-aged participants (40–64 years of age), 2098 were excluded because they did not receive a hearing threshold test or respond to a tinnitus-related questionnaire (*n* = 887), did not provide their chocolate consumption frequency (*n* = 874), or had missing values for confounders (*n* = 337). Finally, 3575 participants from the 2012–2013 KNHANES aged 40 to 65 years were enrolled in the study (Figure 1).

### 2.2. Audiological Assessment

A structured physical examination, to exclude middle and external ear problems, was conducted by a physician. The pathologies of perforation or retraction of the tympanic membrane, otitis media with effusion, and cholesteatoma were identified by ear endoscopy. All subjects underwent pure-tone audiometry for six different octave frequencies (0.5, 1, 2, 3, 4, and 6 kHz) in a soundproof room. The mean hearing threshold was calculated as the average of the hearing thresholds at 0.5, 1, 2, and 4 kHz. The mean high-tone hearing threshold was determined using the average of the hearing thresholds at 3, 4, and 6 kHz. To exclude subjects with systemic diseases, blood samples were collected and analyzed at the Neodin Medical Institute in Seoul, Korea.

With regard to tinnitus, all subjects were also interviewed about the presence of tinnitus and tinnitus-related annoyance. The severity of tinnitus was classified as follows: “no,” “slightly annoying”, and “very annoying and difficult to sleep”. In this study, a response of “slightly annoying” or “very annoying and difficult to sleep” was defined as tinnitus-related annoyance. The parameters of mean hearing threshold according to frequency, presence of tinnitus, and tinnitus-related annoyance were used to evaluate the association with chocolate.

### 2.3. Potential Confounders

The following potential confounders were adjusted for: sleep duration, stress severity, income, current smoking habits, alcohol consumption, noise exposure, and medical conditions. The information obtained included sleep duration (<6, 6–7, 7–8, or ≥8 h), rate of perceived stress, income (<25%, 25–50%, 50–75%, or >75% of the equalized household income per month), current smoking habits, exposure to indoor second-hand smoke (at work or at home), alcohol consumption (social drinker, heavy drinker, or problem drinker), and difficulties in controlling alcohol use. The duration of occupational exposure to noise and earphone and headphone use were also measured. In addition, the health status (presence of hypertension, diabetes, dyslipidemia, anemia, kidney failure, thyroid disorder, and menopause) of the subjects was evaluated. Finally, we used sweet products without cocoa (including ice cream, cake, and cookies) as a confounder in the evaluation of the non-specific effect of pleasant sensations.

### 2.4. Assessment of Chocolate Consumption

As in previous study [19], chocolate consumption was assessed using a food-frequency questionnaire (FFQ). Participants were asked to indicate how frequently they consumed chocolate in the previous year, from 2012 to 2013; there were 10 possible responses (never or seldom, once per month, two to three times per month, once per week, two to four times per week, five to six times per week, once per day, twice per day, three times per day, and no response). In addition, average chocolate intake was categorized into rarely (less than once per month), one quarter of a chocolate tablet, one half of a chocolate tablet, one chocolate tablet, and no response. Information on the frequency and quantity of chocolate consumed over the past year was collected by trained dietitians after the health interview (Appendix A).

### 2.5. Statistical Analysis

All statistical analyses were performed using SAS software (version 9.2; SAS Institute, Cary, NC, USA). The subjects’ demographic and clinical characteristics are presented as medians (interquartile ranges) or numbers (proportions). The comparison between groups, performed by Fisher exact test (binary covariates), chi-squared test (more than three categories), or Wilcoxon rank sum test (continuous covariates), was deemed appropriate. The association between chocolate consumption and hearing loss and tinnitus was evaluated by means of multivariate logistic regression models with adjustment for the following potential confounders: age; sex; sleep duration; perceived stress; current smoking habits; exposure to indoor second-hand smoke; heavy drinking; drinking-related problem; duration of earphone use; occupational exposure to noise; menopause; and history of hypertension, diabetes mellitus, dyslipidemia, anemia, kidney failure, and thyroid disorder. The Spearman correlation rank order was used to evaluate the correlation between the frequency of chocolate consumption and the severity of hearing loss. In the multivariate models of tinnitus or tinnitus-related annoyance, we controlled for unilateral or bilateral hearing loss as well as the above potential confounders. 

## 3. Results

The rate of any hearing loss (unilateral or bilateral hearing loss) was significantly lower in the subjects who consumed chocolate (26.78% (338/1262)) than in those who did not (35.97% (832/2313)) (*p* < 0.001, Table 1). In addition, chocolate consumption decreased the risk of bilateral hearing loss (13.31% (168/1262) vs. 20.32% (470/2313), *p* < 0.001) and high-tone hearing loss (51.58% (651/1262) vs. 63.60% (1,471/2313), *p* < 0.001), respectively.

In a multivariate logistic regression analysis, compared to chocolate non-consumers, the subjects who consumed chocolate had lower odds of any hearing loss (adjusted odds ratio (OR) = 0.83, 95% confidence interval (CI) = 0.70 to 0.98, *p* = 0.03), bilateral hearing loss (adjusted OR = 0.79, 95% CI = 0.64 to 0.98, *p* = 0.03), and high-tone hearing loss (adjusted OR = 0.78, 95% CI = 0.66 to 0.91, *p* = 0.02) (Table 2). 

In contrast to chocolate, neither hearing loss nor bilateral hearing loss was associated with the consumption of sweet products without cocoa, according to multivariable logistic regression model. Furthermore, chocolate was significantly associated with a decrease of hearing loss after adjusting for cofounders, including consumption of sweet products without cocoa (Table 3). 

Moreover, Spearman’s rank order correlation analyses of the data revealed a significant inverse correlation between the mean hearing threshold and the frequency of chocolate consumption per week (*ρ* = −0.117, 95% CI = −0.15 to −0.08, *p* < 0.001) (Figure 2A). In addition, the mean threshold of high-tone frequencies was negatively correlated with the frequency of chocolate consumption per week (*ρ* = −0.121, 95% CI = −0.15 to −0.09, *p* < 0.001) (Figure 2B).

Neither the rate of tinnitus nor the rate of tinnitus-related annoyance differed significantly according to chocolate consumption. In addition, subjects who consumed chocolate had 9% and 11% lower odds of tinnitus (adjusted OR = 0.91, 95% CI = 0.74 to 1.10) and tinnitus-related annoyance (adjusted OR = 0.89, 95% CI = 0.67 to 1.18), respectively (Table 2). However, the association between chocolate consumption and tinnitus and tinnitus-related annoyance was not significant.

## 4. Discussion

To our knowledge, this is the first study to explore the effect of chocolate on hearing loss and tinnitus in a large cohort of middle-aged people. The rate of hearing loss was significantly lower in the subjects who consumed chocolate than in those who did not. Additionally, there was an inverse correlation between the severity of hearing loss and the frequency of chocolate consumption. These results support those of previous animal studies, which reported that various compounds in chocolate protect against hearing loss [20]. Specifically, there was no association between hearing loss and the consumption of sweet products without cocoa. Contrary to our hypothesis, chocolate consumption was not associated with tinnitus or tinnitus-related annoyance.

The cochlea is susceptible to oxidative stress because of its high energy requirements and lack of an adequate collateral blood supply [21]. Thus, the cochlea is likely to be an initial target organ of ischemia. Several clinical risk factors, such as noise exposure and smoking, are associated with oxidative stress in the cochlea [22]. We adjusted for possible confounders but were unable to perform strict matching of confounders due to the high drop-out rate. Our data indicate that chocolate consumption is independently associated with a decreased risk of hearing loss.

Although we could not evaluate the direct effect of chocolate on hearing loss by quantifying cocoa, our results suggest that chocolate may prevent hearing loss by a mechanisms other than the pleasant sensation of chocolate. Those who did not consume chocolate had higher risks of hypertension and dyslipidemia than did those who consumed chocolate. Chocolate exerts antioxidant and anti-inflammatory effects and has been shown to have therapeutic benefits for patients with cardiovascular diseases. Specifically, cocoa, a major ingredient of chocolate, attenuates vascular risks by reducing blood pressure and improving endothelium-dependent vasodilation [23]. In line with this, several investigations demonstrated the causal relationship between vascular risk factors and hearing loss [24,25]. Thus, our results suggest that chocolate decreases the rates of hypertension and dyslipidemia, which enables the preservation of hearing loss. In addition to vascular risk factors, chocolate products provide therapeutic benefits to patients with neurological, intestinal, and neurodegenerative diseases [18].

Chocolate is an important dietary source of flavonoids, a subclass of polyphenols [26]. Polyphenols exert antioxidant and anti-inflammatory effects [20,27]. A recent animal study demonstrated that polyphenols ameliorated oxidative stress inside the cochlea by downregulating the apoptotic signaling pathway [16]. Notably, the level of oxidative stress in the cochlea increased with age; this effect was attenuated by polyphenols. Consistent with this, polyphenols exhibited significant radical scavenging activity in individuals exposed to noise [17] and significantly improved the auditory thresholds in rats [20]. Moreover, chocolate-mediated nitric oxide (NO) release enhances blood circulation in the inner ear and decreases the levels of inflammatory markers; e.g., C-reactive protein, COX-2, and atherogenesis [18,28]. The biological activities of chocolate vary according to the processing strategy used in its production, but chocolate produced using a gentle processing technique may protect against hearing loss, particularly in middle-aged people.

Unexpectedly, chocolate intake was not associated with tinnitus or tinnitus-related annoyance. In humans there is a relationship between the hearing-loss frequency and the tinnitus pitch, suggesting a strong relationship between auditory deafferentation and tinnitus. However, the association between hearing loss and tinnitus may not be straightforward. For example, subjective tinnitus was reported by approximately 19% of individuals with normal hearing in a cohort study in Korea [14], suggesting that subjective tinnitus can develop in the absence of hearing loss. The persistence of tinnitus after cochlear nerve section suggests a central origin [29]. Although the pathophysiological mechanisms of tinnitus are unclear, it may be caused by maladaptive cortical plasticity between auditory and non-auditory regions [11]. Thus, chocolate may improve cochlear rheology, such as microcirculation and vasodilation, but does not significantly ameliorate tinnitus or tinnitus-related annoyance.

The lack of an association between chocolate consumption and tinnitus may be attributable to strict adjustment for comorbidities, e.g., hearing loss, stress, and sleep [30]. Moreover, systemic inflammatory mediators that predispose individuals to tinnitus are likely to be associated with the mechanisms of the protective effect of chocolate [31]. Specifically, chocolate restores imbalances in lipid and glucose levels, which are linked to tinnitus [32,33]. Therefore, other as-yet-unknown effects of chocolate are likely to contribute to the development of tinnitus. This is in line with the borderline significance detected in univariate regression analyses, which was lost after adjustment for confounders.

This study had several limitations that should be noted. First, we could not assess the causality of the relationship between chocolate consumption and hearing loss due to the cross-sectional nature of the study. Moreover, no information on chocolate consumption was available for participants over 65 years of age. Given the protective effect of chocolate on age-related hearing loss, the association might have been significant if older persons had been included. Second, the processing technique used in the production of chocolate affects its total polyphenol content [15]. The data regarding chocolate were obtained using subjective questionnaires; thus, chocolate’s functional properties could not be evaluated. The levels of polyphenols, such as epicatechins and catechins, in plasma are correlated with those of markers of antioxidant activity in humans [34]. Third, the type of chocolate, dose, and duration of consumption were not controlled, and milk chocolate and chocolate drinks reportedly do not exert a significant effect on health [15]. To establish causality, a randomized controlled study of the functional properties of chocolate is needed. Lastly, we were unable to compare the effect of chocolate according to laterality. There is some evidence suggesting that the genotype influences the phenotype, particularly in the laterality, of self-reported tinnitus; the discrepancy between unilateral and bilateral tinnitus was more evident in men than women [35]. Although the effect of chocolate on hearing loss and tinnitus was similar irrespective of sex as documented by interaction and multivariate analyses in this study (Appendix A), this finding awaits further confirmation. 

Nonetheless, the present study had several strengths. First, a large cohort from a nationally representative database was analyzed. Second, this is the first study of the relationship between chocolate intake and hearing loss. Our results will enhance nutritional support for patients with hearing loss.

## 5. Conclusions

Our results suggest that chocolate plays an otoprotective role against hearing loss but not tinnitus in middle-aged people.

## Figures and Tables

**Figure 1 nutrients-11-00746-f001:**
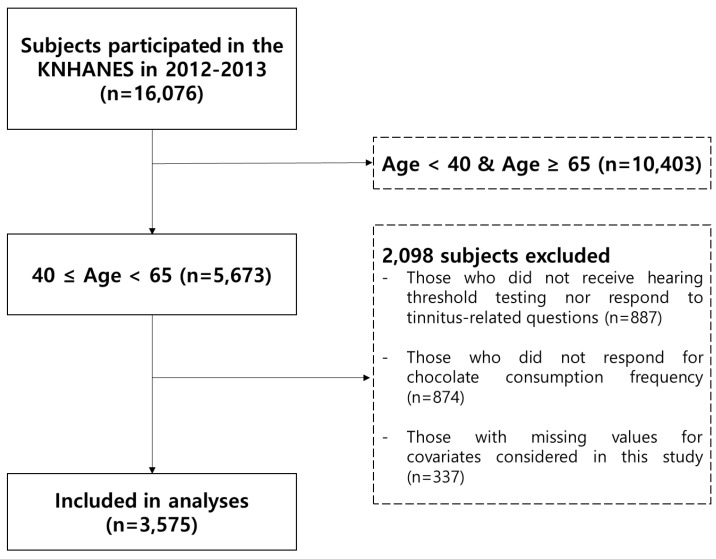
Schematic illustration of the selection of subjects.

**Figure 2 nutrients-11-00746-f002:**
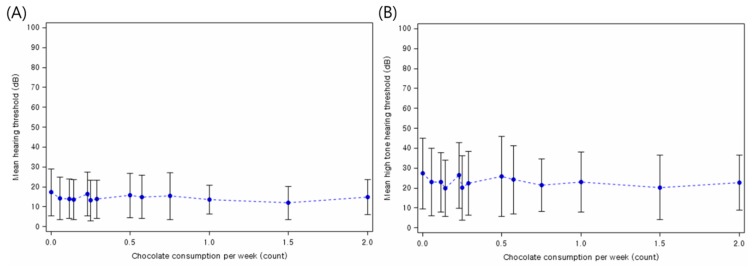
Scatter plot of the severity of hearing loss and the frequency of chocolate consumption per week. Chocolate consumption was inversely correlated with the (**A**) mean hearing threshold (average of 0.5, 1, 2, and 4 kHz) and (**B**) mean high-tone hearing threshold (average of 3, 4, and 6 kHz).

**Table 1 nutrients-11-00746-t001:** Characteristics of the 40–64-year-old subjects according to chocolate consumption.

	Chocolate Consumption			*p*-Value ^1^
	Total (*n* = 3575)	More Than Once (*n* = 1262)	None (*n* = 2313)	
Hearing loss ^2^, *n* (%)				
Unilateral or bilateral	1170 (32.73%)	338 (26.78%)	832 (35.97%)	<0.0001
Bilateral	638 (17.85%)	168 (13.31%)	470 (20.32%)	<0.0001
High-tone hearing loss	2122 (59.36%)	651 (51.58%)	1471 (63.60%)	<0.001
Tinnitus, *n* (%)	811 (22.69%)	266 (21.08%)	545 (23.56%)	0.0947
Tinnitus-related annoyance, *n* (%)	259 (7.24%)	78 (6.18%)	181 (7.83%)	0.0790
Age (year), median (interquartile range, IQR)	52 (45, 58)	50 (44, 56)	53 (47, 59)	<0.0001
Male, *n* (%)	1415 (39.58%)	493 (39.06%)	922 (39.86%)	0.6677
Monthly household income ^3^, median (IQR)	333.33 (200, 518.33)	400 (250, 591.67)	310.67 (184.17, 500)	<0.0001
Use of earphones, *n* (%)	186 (5.20%)	90 (7.13%)	96 (4.15%)	0.0002
Duration of earphone use (min), median (IQR)				
Total	0 (0, 0)	0 (0, 0)	0 (0, 0)	0.0002
User of earphones	60 (30, 60)	30 (20, 60)	60 (30, 90)	0.0617
Occupational exposure to noise, *n* (%)	587 (16.42%)	186 (14.74%)	401 (17.34%)	0.0473
Duration of occupational exposure to noise (months), median (IQR)				
Total	0 (0, 0)	0 (0, 0)	0 (0, 0)	0.0448
Occupational exposure to noise	96 (36, 216)	96 (36, 216)	108 (36, 216)	0.8920
Sleep duration (hours), *n* (%)				0.0295
<6	504 (14.1%)	163 (12.92%)	341 (14.74%)	
6–7	1,039 (29.06%)	377 (29.87%)	662 (28.62%)	
7–8	1,099 (30.74%)	418 (33.12%)	681 (29.44%)	
≥8	933 (26.1%)	304 (24.09%)	629 (27.19%)	
High perceived stress, *n* (%)	770 (21.54%)	274 (21.71%)	496 (21.44%)	0.8648
Exposure to indoor second-hand smoke				
At work, *n* (%)	1162 (32.5%)	418 (33.12%)	744 (32.17%)	0.5753
At home, *n* (%)	349 (9.76%)	123 (9.75%)	226 (9.77%)	>0.9999
Current smoking, *n* (%)	653 (18.27%)	203 (16.09%)	450 (19.46%)	0.0128
Heavy drinking ^4^, *n* (%)	665 (18.60%)	193 (15.29%)	472 (20.41%)	0.0002
Difficulties in controlling alcohol use, *n* (%)	294 (8.22%)	87 (6.89%)	207 (8.95%)	0.0354
Having drinking-related problem in life, *n* (%)	165 (4.62%)	56 (4.44%)	109 (4.71%)	0.7393
Menopause (females)				<0.0001
Yes	1209 (55.97%)	372 (48.37%)	837 (60.17%)	
No	951 (44.03%)	397 (51.63%)	554 (39.83%)	
Hypertension, *n* (%)	683 (19.10%)	196 (15.53%)	487 (21.05%)	<0.0001
Diabetes mellitus, *n* (%)	247 (6.91%)	60 (4.75%)	187 (8.08%)	0.0001
Anemia, *n* (%)	284 (7.94%)	115 (9.11%)	169 (7.31%)	0.0606
Kidney failure, *n* (%)	13 (0.36%)	6 (0.48%)	7 (0.30%)	0.4004
Thyroid disorder, *n* (%)	85 (2.38%)	35 (2.77%)	50 (2.16%)	0.2527
Dyslipidemia, *n* (%)	331 (9.26%)	87 (6.89%)	244 (10.55%)	0.0003

^1^*p*-values by Fisher exact test (binary covariates), chi-squared test (more than three categories) and Wilcoxon rank sum test (continuous covariates). ^2^ Hearing loss ≥20 dB for four frequency average of pure-tone thresholds at 500, 1000, 2000, and 4000 Hz. ^3^ Monthly household income (10,000 Korean won). ^4^ Heavy drinking defined as more than three drinks per average drinking session more than twice a week.

**Table 2 nutrients-11-00746-t002:** Odds ratios (OR) and 95% confidence intervals (CI) for hearing loss and tinnitus according to chocolate consumption.

	*n* = 3575	Univariate Analysis		Multivariate Analysis	
		OR (95% CI)	*p*-Value	OR (95% CI)	*p*-Value
**Hearing loss** **(unilateral or bilateral)**	1170: 2405	0.651 (0.560, 0.757)	<0.0001	0.829 (0.701, 0.980) ^1^	0.0285
**Hearing loss (bilateral)**	638: 2937	0.602 (0.497, 0.729)	<0.0001	0.791 (0.641, 0.976) ^1^	0.0287
**High-tone hearing loss**	2122: 1453	0.610 (0.531, 0.701)	<0.0001	0.777 (0.661, 0.912) ^1^	0.0021
**Tinnitus**	811: 2764	0.866 (0.734, 1.023)	0.0902	0.911 (0.767, 1.081) ^2^	0.2847
**Tinnitus-related annoyance**	259: 3316	0.776 (0.590, 1.021)	0.0705	0.886 (0.668, 1.176) ^2^	0.4036

^1^ Adjusted for: age; sex; perceived stress; exposure to indoor second-hand smoke; current smoking habits; heavy drinking; drinking-related problems; menopause; histories of hypertension, diabetes mellitus, anemia, kidney failure, thyroid disorder, and dyslipidemia; income level; sleep duration; duration of occupational exposure to noise; and earphone and headphone use time. ^2^ Adjusted for all covariates used in the hearing-loss model in addition to hearing loss (unilateral or bilateral).

**Table 3 nutrients-11-00746-t003:** Odds ratios and 95% confidence intervals for hearing loss and tinnitus according to consumption of chocolate and sweet products without cocoa.

	Multivariable Analysis		
	Consumption Per Week (Reference = No)	OR (95% CI)	*p*-Value
**Hearing loss (unilateral or bilateral)**	Chocolate	0.835 (0.703, 0.992) ^1^	0.0406
	Cookie	0.989 (0.973, 1.006) ^1^	0.1912
	Ice cream	1.055 (0.967, 1.151) ^1^	0.2308
	Cake	0.975 (0.861, 1.105) ^1^	0.6931
**Hearing loss (bilateral)**	Chocolate	0.766 (0.617, 0.951) ^1^	0.0156
	Cookie	0.989 (0.968, 1.010) ^1^	0.3043
	Ice cream	1.146 (1.039, 1.265) ^1^	0.0063
	Cake	1.041 (0.901, 1.203) ^1^	0.5826
**Tinnitus**	Chocolate	0.920 (0.772, 1.097) ^2^	0.3536
	Cookie	1.001 (0.985, 1.018) ^2^	0.8901
	Ice cream	1.009 (0.922, 1.104) ^2^	0.8478
	Cake	0.932 (0.814, 1.067) ^2^	0.3081
**Tinnitus-related annoyance**	Chocolate	0.876 (0.655, 1.172) ^2^	0.3735
	Cookie	0.987 (0.956, 1.020) ^2^	0.4431
	Ice cream	1.120 (0.990, 1.267) ^2^	0.0721
	Cake	0.973 (0.787, 1.204) ^2^	0.8028

^1^ Variables included in multivariable models: age; sex; perceived stress; exposure to indoor second-hand smoke; current smoking habits; heavy drinking; drinking-related problem; menopause; history of hypertension, diabetes mellitus, anemia, kidney failure, thyroid disorder, and dyslipidemia; income level; sleep duration; duration of occupational exposure to noise; and earphone and headphone use time; chocolate consumption; cookie consumption; ice cream consumption; cake consumption. ^2^ Variables included in multivariable models: all covariates used in the hearing-loss model as well as hearing loss (unilateral or bilateral).

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
