# Peer review of "Association of Chocolate Consumption with Hearing Loss and Tinnitus in Middle-Aged People Based on the Korean National Health and Nutrition Examination Survey 2012–2013"

_nutrients, 2019, doi:10.3390/nu11040746_

Reviewer 1 Report

This is a well-design study that shows that chocolate consumption could protect against hearing loss in 40–64 years old in a study including 3575 individuals. This effect was also found for high frequency hearing loss. Of note, no effect was noted for tinnitus.

The methods are clearly detailed and the conclusions are based on the results and this reviewer considers that study could be potentially accepted. I would appreciate that the authors could clarify some issues.

Questions:

1. Could the author compare the effect of chocolate between unilateral and bilateral tinnitus?

2. In Table 1, the groups are different in 2 vascular risk factors: hypertension and dyslipidaemia. This could be related with the effect of chocolate and mentioned in th discussion.

3. I also wish that the authors could analyzed the data by gender to show if the effect of chocolate has a different effect on hearing thresholds and tinnitus in men and women. There is some evidence of sexual dimorfism in hearing loss and tinnitus in twins studies with tinnitus (Maas et al.2017)

Suggested reference

1. Maas IL, Brüggemann P, Requena T, Bulla J, Edvall NK, Hjelmborg JVB, et al. Genetic susceptibility to bilateral tinnitus in a Swedish twin cohort. Genet Med. 2017 Sep;19(9):1007–12.

Author Response

RESPONSES TO THE REVIEWERS’ COMMENTS:

First, we appreciate all of your valuable comments and suggestions. All revisions to the manuscript are shown in blue and italics for the reviewers’ convenience.

Reviewer’s comments

#1 reviewer :

1. Could the author compare the effect of chocolate between unilateral and bilateral tinnitus?

[Response] Regrettably, we were unable to compare the effect of chocolate according to tinnitus laterality. With regard to tinnitus, the subjects were asked about the presence and tinnitus-related annoyance. As per the reviewer’s suggestion, we added this limitation in the discussion section.

2. In Table 1, the groups are different in 2 vascular risk factors: hypertension and dyslipidemia. This could be related with the effect of chocolate and mentioned in discussion.

[Response] Thank you for providing us the opportunity to clarify this important issue. As shown in Table 1, the group without chocolate consumption had a higher risk of hypertension and dyslipidemia than the group with chocolate.

Chocolate that exerts antioxidant and anti-inflammatory effects has shown to have therapeutic benefits from patients with cardiovascular diseases. Specifically, Cocoa, a major ingredient of chocolate, attenuates the vascular risk by reducing blood pressure and improving endothelium-dependent vasodilation [1]. In line with this, several investigations demonstrated the causal relationship between vascular risk factors and hearing loss [2,3].  

Thus, our results suggest that chocolate decrease the rate of hypertension and dyslipidemia, which enables the preservation of hearing loss.

3. I also wish that the authors could analyzed the data by gender to show if the effect of chocolate has a different effect on hearing thresholds and tinnitus in men and women. There is some evidence of sexual dimorfism in hearing loss and tinnitus in twins studies with tinnitus (Maas et al.2017)

1. Maas IL, Brüggemann P, Requena T, Bulla J, Edvall NK, Hjelmborg JVB, et al. Genetic susceptibility to bilateral tinnitus in a Swedish twin cohort. Genet Med. 2017 Sep;19(9):1007–12.

[Response] Thank you for providing us the opportunity to clarify this critical issue. Also, we appreciate for providing a reference.

1) According to the reference., Mass et al. suggested that genotype could influence the tinnitus (self-reported tinnitus) phenotype, especially for laterality. The discrepancy between unilateral and bilateral tinnitus was more evident in men than women [4].

2) As per the reviewer’s suggestion, we analyzed whether the effects of chocolate on hearing loss and tinnitus were different according to sex (supplement table)

With regard to the interaction effect, no statistical significance was observed regardless of sex. Moreover, the effect of chocolate on hearing loss and tinnitus (estimate) was similar based on multivariable analyses according to sex.

chocolate   interaction

Any hearing loss

(unilateral   or bilateral)

0.8822

Hearing   loss (bilateral)

0.0567

Tinnitus

0.2667

Tinnitus-related   annoyance

0.4143

The interaction between sex and chocolate consumption on each of hearing loss and tinnitus outcomes was tested using an interaction term in a multivariable logistic regression model with all the potential confounders considered in this study.

3) Although some evidences of sexual dysmorphism in hearing loss and tinnitus [4], the effect of chocolate on hearing loss and tinnitus was similarly regardless of sex, as documented by interaction and multivariable analyses.

Estimate

95% CI

Lower   Upper

P-value

Estimate

95% CI

Lower   Upper

P-value

Men

Any   hearing loss

(unilateral   or bilateral)

0.652

0.521

0.817

0.0002

0.852

0.662

1.098

0.2158

Hearing   loss (bilateral)

0.545

0.418

0.712

<.0001< span="">

0.719

0.536

0.965

0.0282

Tinnitus

0.844

0.64

1.113

0.23

0.924

0.691

1.237

0.5973

Tinnitus-related   annoyance

0.57

0.355

0.914

0.0196

0.709

0.432

1.166

0.1753

Women

Any   hearing loss

(unilateral   or bilateral)

0.642

0.521

0.791

<.0001< span="">

0.823

0.656

1.033

0.0938

Hearing   loss (bilateral)

0.665

0.502

0.88

0.0044

0.886

0.654

1.2

0.434

Tinnitus

0.876

0.712

1.078

0.2123

0.902

0.728

1.117

0.3438

Tinnitus-related   annoyance

0.925

0.658

1.3

0.6538

1.022

0.719

1.452

0.9042

References

1.         Grassi, D.; Necozione, S.; Lippi, C.; Croce, G.; Valeri, L.; Pasqualetti, P.; Desideri, G.; Blumberg, J.B.; Ferri, C. Cocoa reduces blood pressure and insulin resistance and improves endothelium-dependent vasodilation in hypertensives. Hypertension 2005, 46, 398-405, doi:10.1161/01.HYP.0000174990.46027.70.

2.         Bener, A.; Al-Hamaq, A.; Abdulhadi, K.; Salahaldin, A.H.; Gansan, L. Interaction between diabetes mellitus and hypertension on risk of hearing loss in highly endogamous population. Diabetes Metab Syndr 2017, 11 Suppl 1, S45-S51, doi:10.1016/j.dsx.2016.09.004.

3.         Lin, B.M.; Curhan, S.G.; Wang, M.; Eavey, R.; Stankovic, K.M.; Curhan, G.C. Hypertension, Diuretic Use, and Risk of Hearing Loss. Am J Med 2016, 129, 416-422, doi:10.1016/j.amjmed.2015.11.014.

4.         Maas, I.L.; Bruggemann, P.; Requena, T.; Bulla, J.; Edvall, N.K.; Hjelmborg, J.V.B.; Szczepek, A.J.; Canlon, B.; Mazurek, B.; Lopez-Escamez, J.A., et al. Genetic susceptibility to bilateral tinnitus in a Swedish twin cohort. Genet Med 2017, 19, 1007-1012, doi:10.1038/gim.2017.4.

5.         Wissler, C. The Spearman Correlation Formula. Science 1905, 22, 309-311, doi:10.1126/science.22.558.309.

Reviewer 2 Report

In this article, the authors report the effect of the frequency of chocolate consumption on hearing loss. They also assessed the effect on tinnitus, which was not found.

My main concern is the abscence of the link with the percentage of cacao in the chocolate - as the authors admit, they did not take the type of chocolate into account. However, the percentage of cocoa can be several times different between the types of cholocate. If the percent of cacao is highly variable, how it is possible to define the effect of chocolate? It can be a non-specific effect of pleasant sensations, which can also have the protective effect at the molecular level. 

If the percentage of cacao is inavailable in the database, a control of a sweet product without cocoa can be used (e.g. caramel and other non-chocolate candies). Otherwise, I doubt that the chocolate-specific effect is reported.

In Figure 2 the observed effects are not clear. I would like to see the means, standard deviations and a fitted curve. Perhaps, a non-linear fit would provide a better visualisation of the effect.

Author Response

RESPONSES TO THE REVIEWERS’ COMMENTS:

First, we appreciate all of your valuable comments and suggestions. All revisions to the manuscript are shown in blue and italics for the reviewers’ convenience.

Reviewer’s comments

#2 reviewer :

1. My main concern is the absence of the link with the percentage of cacao in the chocolate - as the authors admit, they did not take the type of chocolate into account. However, the percentage of cocoa can be several times different between the types of cholocate. If the percent of cacao is highly variable, how it is possible to define the effect of chocolate? It can be a non-specific effect of pleasant sensations, which can also have the protective effect at the molecular level. If the percentage of cacao is inavailable in the database, a control of a sweet product without cocoa can be used (e.g. caramel and other non-chocolate candies). Otherwise, I doubt that the chocolate-specific effect is reported.

[Response] We wish to express our gratitude to you for helpful comments. We, sincerely, agree with your opinion.

1) Regrettably, we were unable to retrieve the information about the percentage of cacao between the types of chocolate. We have added this limitation in the discussion section.

2) As per the reviewer’ suggestion, we analyzed the effect of a sweet product without cocoa (including ice cream, cake, and cookie) on hearing loss and tinnitus. In multivariable linear analyses after adjusting confounder, in contrast to chocolate, either any hearing loss or bilateral hearing loss were not associated with quantitation of sweet products without cocoa. Furthermore, the chocolate was significantly associated with a decrease of hearing loss after further adjusting cofounders combined with sweet products without cocoa. These results were consistent with their association according to the presence of consumption. 

3) Although we could not reveal the direct effect of chocolate on hearing loss via quantitation of cocoa, our results suggest that chocolate may act to prevent hearing loss by potential  working mechanisms other than the pleasant sensation of chocolate. 

Table 3. Odds ratios and 95% confidence intervals for hearing loss and tinnitus according to consumption of chocolate and sweet products without cocoa

Multivariable analysis

Consumption per week   (reference=no)

OR (95% CI)

P-value

Hearing loss (unilateral or   bilateral)

Chocolate

0.835   (0.703, 0.992)1)

0.0406

Cookie

0.989   (0.973, 1.006)1)

0.1912

Ice cream

1.055   (0.967, 1.151)1)

0.2308

Cake

0.975   (0.861, 1.105)1)

0.6931

Hearing loss (bilateral)

Chocolate

0.766   (0.617, 0.951)1)

0.0156

Cookie

0.989 (0.968,   1.010)1)

0.3043

Ice cream

1.146   (1.039, 1.265)1)

0.0063

Cake

1.041   (0.901, 1.203)1)

0.5826

Tinnitus

Chocolate

0.920   (0.772, 1.097)2)

0.3536

Cookie

1.001   (0.985, 1.018)2)

0.8901

Ice cream

1.009   (0.922, 1.104)2)

0.8478

Cake

0.932   (0.814, 1.067)2)

0.3081

Tinnitus-related annoyance

Chocolate

0.876   (0.655, 1.172)2)

0.3735

Cookie

0.987   (0.956, 1.020)2)

0.4431

Ice cream

1.120   (0.990, 1.267)2)

0.0721

Cake

0.973   (0.787, 1.204)2)

0.8028

1) Variables included in multivariable models: age; sex; perceived stress; exposure to indoor second-hand smoke; current smoking; heavy drinking; drinking-related problem; menopause; history of hypertension, diabetes mellitus, anaemia, kidney failure, thyroid disorder, and dyslipidaemia; income level; sleep duration; duration of occupational exposure to noise; and earphones and headphone use time; chocolate consumption; cookie consumption amount; ice cream consumption amount; cake consumption amount.

2) Variables included in multivariable models: all covariates used in the hearing-loss model as well as hearing loss (unilateral or bilateral).

2. In Figure 2 the observed effects are not clear. I would like to see the means, standard deviations and a fitted curve. Perhaps, a non-linear fit would provide a better visualisation of the effect.

[Response] We appreciate the reviewer’s valuable comment.

1) As your recommendation, we inserted the plots of means and standard deviations for the frequency of chocolate consumption per week.

2) For these plots, due to low frequencies of more than once a week (seen in Figure 2),

the data of more than once a week were grouped to two groups of twice or more than twice a week (2) and more than once but less than twice a week (1< frequencies<2) which were coded as 2 and 1.5.

3) In regards to a fitted curve, the Spearman correlation based on the ranks of data were employed to explore the associations between the frequency of chocolate consumption per week and hearing threshold. Typically, the Spearman correlation was used if the distribution of the chocolate consumption was not normally distributed and the frequency was obtained from the questionnaire with 10 possible responses (never or seldom, once per month, two to three times per month, once per week, two to four times per week, five to six times per week, once per day, twice per day, three times per day) [5].

4) Instead of a fitted curve, the line connecting means have added to plots of the means for a better visualization of the effect.

 References

5.      Wissler, C. The Spearman Correlation Formula. Science 1905, 22, 309-311, doi:10.1126/science.22.558.309.

Round  2

Reviewer 2 Report

I am satisfied with this revision.